# Submicron-Sized Nb-Doped Lithium Garnet for High Ionic Conductivity Solid Electrolyte and Performance of Quasi-Solid-State Lithium Battery

**DOI:** 10.3390/ma13030560

**Published:** 2020-01-24

**Authors:** Yan Ji, Cankai Zhou, Feng Lin, Bingjing Li, Feifan Yang, Huali Zhu, Junfei Duan, Zhaoyong Chen

**Affiliations:** 1College of Materials Science and Engineering, Changsha University of Science and Technology, Changsha 410114, China; juefly@stu.csust.edu.cn (Y.J.); zhoucankai@stu.csust.edu.cn (C.Z.); 18216359528@163.com (F.L.); krystalbingjingli@163.com (B.L.); yff_0413@126.com (F.Y.); junfei_duan@csust.edu.cn (J.D.); 2College of Physics and Electronic Science, Changsha University of Science and Technology, Changsha 410114, China; juliezhu2005@126.com

**Keywords:** solid-state electrolyte, submicron powder, garnet, lithium-ion conductivity, solid-state batteries

## Abstract

The garnet Li_7_La_3_Zr_2_O_12_ (LLZO) has been widely investigated because of its high conductivity, wide electrochemical window, and chemical stability with regards to lithium metal. However, the usual preparation process of LLZO requires high-temperature sintering for a long time and a lot of mother powder to compensate for lithium evaporation. In this study submicron Li_6.6_La_3_Zr_1.6_Nb_0.4_O_12_ (LLZNO) powder―which has a stable cubic phase and high sintering activity―was prepared using the conventional solid-state reaction and the attrition milling process, and Li stoichiometric LLZNO ceramics were obtained by sintering this powder―which is difficult to control under high sintering temperatures and when sintered for a long time―at a relatively low temperature or for a short amount of time. The particle-size distribution, phase structure, microstructure, distribution of elements, total ionic conductivity, relative density, and activation energy of the submicron LLZNO powder and the LLZNO ceramics were tested and analyzed using laser diffraction particle-size analyzer (LD), X-Ray Diffraction (XRD), Scanning Electron Microscope (SEM), Electrochemical Impedance Spectroscopy (EIS), and the Archimedean method. The total ionic conductivity of samples sintered at 1200 °C for 30 min was 5.09 × 10^−4^ S·cm^−1^, the activation energy was 0.311 eV, and the relative density was 87.3%. When the samples were sintered at 1150 °C for 60 min the total ionic conductivity was 3.49 × 10^−4^ S·cm^−1^, the activation energy was 0.316 eV, and the relative density was 90.4%. At the same time, quasi-solid-state batteries were assembled with LiMn_2_O_4_ as the positive electrode and submicron LLZNO powder as the solid-state electrolyte. After 50 cycles, the discharge specific capacity was 105.5 mAh/g and the columbic efficiency was above 95%.

## 1. Introduction

Currently, lithium-ion batteries are widely used in electric vehicles (EVs), hybrid electric vehicles (HEVs), computers, smart grids, wearable devices, etc. [1]. Traditional lithium-ion batteries have organic liquid electrolytes which easily burn and explode under abusive conditions. In addition, with the development of modern society, lithium-ion batteries have gradually moved toward high specific energy. Researchers have studied cathode materials, such as LiCoO_2_, LiMn_2_O_4_, LiFePO_4_, LiNi_0.6_Co_0.2_Mn_0.2_O_2_, LiNi_0.8_Co_0.1_Mn_0.1_O_2_, LiNi_0.8_Co_0.15_Al_0.05_O_2_, and xLi_2_MnO_3_·(1 − x)LiMO_2_, to increase the energy density of lithium-ion batteries [2,3,4,5,6,7,8]. The energy density can be improved by increasing the charging voltage, which will lead to serious side reactions and safety issues. In order to solve the safety problem of lithium-ion batteries, researchers have turned their attention to all-solid-state lithium batteries, which use inorganic electrolytes. The non-flammability, long cycling life and wide electrochemical window of all-solid-state lithium batteries are considered to provide the high safety and high energy density of the next-generation energy storage systems [9,10].

A solid electrolyte is an important component of all-solid-state batteries. It can not only be used as a lithium ionic conductor, substituting for a liquid organic electrolyte, but also can be used to block direct contact between the positive and negative electrodes, like a separator [11]. Solid electrolytes generally contain Li_3_N, LiPON, perovskite, LISICON, NASICON, garnet, etc. [12,13,14,15,16,17]. Some of these solid electrolytes have high ionic conductivity (~10^−3^ S·cm^−1^). However, some issues still exist, such as instability in an ambient atmosphere (Li_10_GeP_2_S_12_, LGPS) and the metal cation being easily reduced by lithium (such as Ti^4+^ in Li_x_La_2/3 -x/3_TiO_3_, LLTO) [18,19]. The cubic garnet LLZO was discovered by Murugan et al. [17] in 2007 and attracted world-wide attention for its advantages, e.g., the simple preparation process, high ionic conductivity (~10^−3^ S·cm^−1^) at room temperature, high electrochemical window (0~6 V vs. Li/Li^+^), and electrochemical stability of lithium metal. On the other hand, LLZO also has some defects, such as an unstable cubic phase and a low density of ceramics [20,21]. Moreover, a mass of LLZO mother powder is needed to compensate for lithium loss when sintering at high temperatures [21,22]. Many solutions have been adopted to solve the above issues. For example, Al, Ga, Fe, Ta, Nb, W, Y, and Sb doping were used to stabilize the cubic phase [23,24,25,26,27,28,29,30]; hot pressing sintering, plasma sintering, and microwave sintering were adopted to improve the relative density and sintering additives [31,32,33]; and Y_2_O_3_, Al_2_O_3_, B_2_O_3_, CaO, MgO, Li_3_PO_4_, and Li_4_SiO_4_ were investigated to reduce the grain-boundary resistance [34,35,36,37,38,39,40]. Usually, in order to evaluate the electrochemical performance, LLZO was used as solid electrolyte in all-solid-state batteries [41,42,43].

In this study, submicron LLZNO powder with a stable cubic phase was synthesized using the conventional solid-state reaction and prepared by the attrition milling process. The submicron LLZNO powder had a high sintering activity, which promoted the sintering process, reduced the sintering temperature and time, and reduced the loss of Li during high-temperature sintering. All these characteristics favored lithium stoichiometry and ionic conductivity. Furthermore, LLZNO ceramics were obtained without mother powder while sintering under reduced temperature and time. The particles-size distribution, phase structure, microtopography, total ionic conductivity, relative density, and activation energy were characterized and analyzed. The quasi-solid-state lithium batteries with LiMn_2_O_4_ as the positive electrode and submicron LLZNO powder as the solid electrolyte were assembled and the electrochemical performance are tested and analyzed.

## 2. Materials and Methods

### 2.1. The Synthesis of LLZNO Powder and Ceramics

A process flow chart of the preparation of submicron LLZNO powder and the sintering of LLZNO ceramics is showed in Figure 1. LLZNO powder was synthesized by the conventional solid-state reaction [44]. Lithium hydroxide monohydrate (LiOH·H_2_O, 98%, Xilong Scientific Co., Ltd., Shantou, China), lanthanum oxide (La_2_O_3_, 99.99%, Shanghai Aladdin Bio-Chem Technology Co., Ltd., Shanghai, China), zirconia (ZrO_2_, 99%, Shanghai Aladdin Bio-Chem Technology Co., Ltd., Shanghai, China), and niobium oxide (Nb_2_O_5_, 99.99%, Sinopharm Chemical Reagent Co., Ltd., Shanghai, China) were used as the raw materials and 10 wt% excess of LiOH·H_2_O was added to compensate for the lithium loss in the high-temperature calcination and sintering process. Yttrium stabilized zirconia (YSZ, 4~8 mm in diameter) and isopropanol (IPA) were used as the ball-grinding medium. The ratio of raw material to grinding balls was 1:5 and the mixed raw material powder was wet-ball ground at 800 rpm in the planetary ball mill for 6 h. The mixture was dried at 70 °C for 14 h, then calcined at 950 °C for 12 h in an alumina crucible with ambient air to obtain the cubic-phase LLZNO powder. LLZNO slurry was attrition milled (Shanghai ROOT mechanical and electrical equipment Co., Ltd., Shanghai, China, 0.7 L volume, 70% filling rate) at 1000 rpm for 2 h, taking YSZ (0.4 mm in diameter) and IPA as the grinding medium, and the solid‒liquid ratio was 1:5. The LLZNO slurry was dried at 70 °C for 14 h to obtain submicron LLZNO powder, from which green pellets (mass of 3 g, 19 mm in diameter and a thickness of about 4 mm) were pressed at 200 MPa under a cold uniaxial press. After that, the green pellets were sintered in a muffle furnace (Changsha Yuandong Electric Furnace Factory) without mother powder at 1100–1200 °C for 30–360 min and then cooled down naturally. At the same time, the green pellets were put on a platinum wire and placed in a crucible of MgO with the lid on to prevent impurity migration and a large amount of volatilization of lithium during the process of high-temperature sintering. For further testing, LLZNO ceramic pellets were polished with 400 and 1000 mesh sandpaper.

### 2.2. Fabrication of Composite Cathodes and Assembly of Quasi-Solid-State Batteries

In order to test the electrochemical performance of submicron LLZNO powder, we prepared a composite cathode and assembled quasi-solid-state batteries. The composite cathode consisted of a LiMn_2_O_4_ positive electrode layer and a submicron LLZNO electrolyte layer. The positive electrode was fabricated by coating the slurry of a mixture containing LiMn_2_O_4_ powder, submicron LLZNO powder, acetylene black (Shanghai Hersbit Chemical Co., Ltd., Shanghai, China), and polyvinylidene difluoride (PVDF, FR905, Shanghai San ai fu New Material Technology Co., Ltd., Shanghai, China), with a weight ratio of 7:2:1:1, onto circular aluminum foils (thickness of 20 µm, Shenzhen Kejingstar Technology Ltd., Shenzhen, China) as the current collector, and the positive material loading was 1.66 mg/cm^2^. Then the composite cathode was fabricated by coating the slurry of a mixture containing submicron LLZNO powder and Polyvinylidene Fluoride (PVDF) with a weight ratio of 9:1 onto the positive electrode layer. The composite cathode was punched into disks with 18 mm diameters after compacted by a roller press (Shenzhen Kejingstar Technology Ltd., Shenzhen, China), and the density of composite cathode was about 2.5 g/cm^3^. Quasi-solid-state batteries were assembled with two electrode coin cells (type CR-2025) in a glove box filled with argon and with lithium metal foil (15 mm in diameter and 1 mm thick, Shenzhen Kejingstar Technology Ltd., Shenzhen, China) as the negative current collector. In addition, 20 µL of a liquid organic electrolyte (1 M LiPF_6_ dissolved in ethyl carbonate (EC) and dimethyl carbonate (DMC) with a ratio of 1:1, CAPCHEM, Shenzhen, China [45]) was added to improve the contact and reduce the interface impedance between the submicron LLZNO electrolyte layer and the anode/cathode [46,47]. Compared with lithium-ion batteries, the added amount of liquid organic electrolyte was small [48,49].

### 2.3. Characterization

X-ray diffraction (XRD, Cu-Kα radiation, λ = 1.542 Å, Bruker D8 ADVANCE, Bruker AXS GmbH, Karlsruhe, Germany) was used to determine the phase of the ceramics pellets at room temperature within 10–60 °C with steps of 5 °C/min. Jade Software was used to match and analyze the phase of the sample. The relative density of ceramics was measured by Archimedes’ method and deionized water was used as the immersion medium. Meanwhile, the theoretical density of LLZNO, calculated by the Jade Software, was 5.20 g/cm^3^, and the relative density was the measured density divided by the theoretical density. The particle size and distribution of the powder were determined by the laser diffraction particle-size test method (LD, Mastersizer 3000, Malvern Instruments Limited, Malvern, UK), and the relative density, refractive index, and absorption rate of the LLZNO powder was 5.20 g/cm^3^, 1.4, and 0.1, respectively. The microtopography of the submicron LLZNO powder and cross section of the ceramic pellets was observed by scanning electron microscope (SEM, TESCAN MIRA3 LMU, TESCAN Orsay Holding, a. s., Brno, Czech Republic). Energy dispersive spectrometer (Oxford X-ray Max20, Oxford Instruments plc, Oxford, UK) mapping was used to characterize the distribution of each element in the cross section of the ceramic pellets. The total lithium ion conductivity of the ceramic pellets was measured by an Electrochemical Impedance Spectroscopy (EIS, Gamry Reference 600+, Gamry Instruments, Warminster, PA, USA) within a temperature range of 25–80 °C, within the frequency of 10 Hz–5 MHz, and with an AC amplitude of 40 mV. The blocking electrode was uniformly coated by a thin silver layer on both sides of the ceramic pellets. The activation energy of the ceramic pellets was measured within a temperature range of 25–80 °C and calculated based on the Arrhenius equation [16]. The quasi-solid-state batteries were tested under the battery charge‒discharge tester (BTS-5V3A, Neware Technology Co., Ltd., Shenzhen, China) at 25 °C, and current density was 0.02 mA/cm^2^.

## 3. Results and Discussions

The XRD pattern of the LLZNO powder is shown in Figure 2b and was identified as cubic phase (PDF 63-0174). The LD result and SEM image of the LLZNO powder after the attrition milling process, which demonstrated a submicron powder, are showed in Figure 2a and Table 1. The D_(10)_, D_(50)_, D_(90),_ and primary particle size of the submicron LLZNO powder were 0.43 µm, 0.59 µm, 0.812 µm, and about 0.1 µm, respectively. The value of D_(3,2)_ (0.575 µm) is similar to that of D_(4,3)_ (0.607 µm), which indicates that the prepared powder had a uniform particle-size distribution. In addition, the powder also had a higher specific surface area (2007 m^2^/kg), which means that the powder had a high sintering activity, which can promote crystal growth and the rapid densification of ceramics in the sintering process.

The XRD patterns of the LLZNO ceramic samples are showed in Figure 3. The phases of all the prepared ceramic samples were identified as cubic phases (PDF 63-0174). The crystal parameters of the different samples are showed in Table 2. The XRD patterns of the samples sintered at 1200 °C × 60 min (SL1) and at 1100 °C × 360 min (SL5) showed a few impure phase peaks, mainly belonging to LiNbO_3_ (PDF 82-0459) and Li_7_NbO_6_ (PDF 29-0816), and due to the decomposition from the high sintering activity of the LLZNO after having been sintered for too long at a high temperature. Moreover, these impure phases decreased the total ionic conductivity of LLZNO ceramics by increasing the resistance of the grain boundary.

AC impedance plots and the enlargement of the LLZNO ceramic pellets under different sintering conditions are showed in Figure 4a, b. The fitting curve of the sample sintered at 1200 °C for 30 min (SL2) is showed in Figure 4c, and it consists of a quasi-semicircle at high frequency and a long diffusion tail at low frequency. The equivalent circuit model R_b_(R_gb_Q_gb_)(R_el_Q_el_), in which R_b_, R_gb_, and R_el_ are resistances originating from the bulk, grain boundaries, and Ag electrodes, is used to fit the plots and is shown in Figure 4d. The total ionic conductivity of the ceramics is mainly decided by R_b_ plus R_gb_. The total ionic conductivity and relative density of the LLZNO ceramic pellets are showed in Figure 4e and Table 2. The highest total ionic conductivity (5.09 × 10^−4^ S·cm^−1^) of the LLZNO ceramic pellets was obtained when sintered at a high temperature and for a short time (SL2, 1200 °C × 30 min), and its relative density is 87.3%. This indicates that high-performance LLZNO ceramics are obtained when sintered at high temperatures only for short sintering times. However, the total ionic conductivity and relative density of ceramic pellets decreased and impure phases occurred when the sintering time was prolonged at 1100 and 1200 °C. The lowest total ionic conductivity (0.35 × 10^−4^ S·cm^−1^) and relative density (83.4%) were obtained when the ceramic pellets were sintered at 1100 °C for 360 min (SL5). Meanwhile, a higher total ionic conductivity (3.49 × 10^−4^ S·cm^−1^) and a higher relative density (90.3%) of ceramic pellets (SL3, 1150 °C × 60 min) were obtained when sintered for 60 min from 1100 to 1200 °C, and this result indicates that, in this study, increasing the sintering temperature too much was disadvantageous for obtaining LLZNO ceramics with good performance.

Arrhenius plots and the linear fitting curve are showed in Figure 5a. The activation energy of ceramics samples is showed in Figure 5b and Table 2, and their values are within the range of 0.31–0.33 eV. This indicates that there was no obvious effect on the activation energy of the ceramics when the green pellets prepared from the submicron LLZNO powder were sintered. The variation tendency of the activation energy was similar to the total ionic conductivity, and the lowest and the highest activation energy was 0.311 eV (SL2, 1200 °C × 30 min) and 0.328 eV (SL5, 1100 °C × 360 min).

SEM images of cross sections of the LLZNO ceramics, which were sintered under different conditions, are showed in Figure 6a–e. We found that the grain size of the ceramics that were sintered for 60 min within a temperature range of 1100 to 1200 °C gradually increased from 1~5 μm (SL4, 1100 °C for 60 min, Figure 6d). A few of the grains were 5 μm and most of the grains were 100~200 μm (SL3, 1150 °C for 60 min, Figure 6c), and finally, all grains were about 200 μm (SL1, 1200 °C for 60 min, Figure 6a). Here, we found a mass of abnormal growth grains (AGGs) [50], as shown in Figure 6a,c,e, and a mass of pores were distributed in the AGGs. Meanwhile, the total ionic conductivity was lower when the AGGs were bigger. This was due to the submicron LLZNO powder having a high sintering activity, which made the crystal grain of the LLZNO ceramics have a high specific surface energy during the high-temperature sintering process, and promoted rapid grain growth and ceramic densification in the sintering process. For the above reasons, the growth rate of the grains was higher than the migration rate of the pores at the grain boundaries when the sintering temperature was higher and the sintering time was longer and the pores could not be discharged from the grain boundaries and finally stay on the inside of the AGGs. As a result, the bulk impedance of the crystal grains increased, and the total ion conductivity was reduced. However, although the submicron LLZNO powder had high sintering activity, the growth of grains could not be entirely promoted in a shorter sintering time and at a lower temperature. Therefore, a mass of grains which stayed in the initial state are shown in Figure 6d (SL4, 1100 °C × 60 min), and this was disadvantageous for lithium-ionic conduction due to the incomplete surface of the LLZNO grains after the attrition milling process. Eventually, the ceramic pellets showed a lower total ionic conductivity (0.51 × 10^−4^ S·cm^−1^). A cross-sectional SEM image of the sample sintered at 1200 °C for 30 min (SL2) is showed in Figure 6b. It was found that the grains grew uniformly (~4 μm), their surfaces were smooth without pores, and they bond tightly with other grains. A highest ionic conductivity of 5.09 × 10^−4^ S·cm^−1^ was obtained, which indicates that the submicron LLZNO powder had a higher sintering activity and high total ionic conductivity LLZNO ceramic pellets could be obtained by sintered at a high temperature for only a short time. At the same time, the LLZNO ceramic pellets which had a higher total ionic conductivity could also be also obtained when the sintering temperature was properly reduced.

Figure 6f shows the SEM image and its EDS mapping, including La, Zr, and Nb in the cross section of the LLZNO ceramic of 1200 °C × 30 min (SL2). The cross section of the sample exhibits a transgranular fracture and an intergranular fracture, and the elements of La, Zr, and Nb are relatively uniformly distributed, which indicates that the Nb element was successfully incorporated into the LLZO lattice. This is also verified by the XRD result. However, the non-uniform distribution of Zr, La, and Nb exists in the central part of the EDS mapping. This indicates that during high-temperature sintering element segregation and depletion occurred due to the different migration rates of the elements. 

The specific capacity and coulombic efficiency of quasi-solid-state batteries with LiMn_2_O_4_ as the positive electrode after 50 cycles of a galvanostatic charge‒discharge test at 25 °C are showed in Figure 7a. The 1st, 2nd, 10th, 20th, and 50th galvanostatic charge‒discharge curves of quasi-solid-state batteries are showed in Figure 7b. The quasi-solid-state batteries showed good cycling performance at a current density of 0.02 mA/cm^2^ and a voltage within 3.0–4.3 V. The first discharge specific capacity was 106.4 mAh/g and the coulomb efficiency was 93.23%. The 2nd, 10th, 20th, and 50th discharge specific capacities were 106.8 mAh/g, 105.3 mAh/g, 106.9 mAh/g, and 105.5 mAh/g, respectively. After 50 cycles of the galvanostatic charge‒discharge test, the coulomb efficiency was maintained at about 95% and the capacity retention rate was 99.15%. The capacity of the batteries increased in the early stage of the galvanostatic charge‒discharge test, which may be caused by the activation of positive material. This indicates that submicron LLZNO powder can be used in quasi-solid-state batteries, and that the specific capacity and the cycling stability of quasi-solid-state batteries are relatively good. Here, the electrochemical performance of quasi-solid-state batteries using submicron LLZNO powder is only discussed, and further research will be carried out in the future.

## 4. Conclusions

In this study, we synthesized Nb-doped stabilized cubic-phase LLZO powder using the conventional solid-state reaction and prepared submicron LLZNO powder using the attrition milling process. Electrolyte ceramics prepared using submicron LLZNO powder can be sintered without mother powder, which reduces the sintering temperature and shortens the sintering time. After being sintered at 1150 °C for 60 min, the total ionic conductivity, relative density, and activation energy was 3.49 × 10^−4^ S·cm^−1^, 90.4%, and 0.316 eV, respectively. When sintered at 1200 °C for 30 min, we obtained the highest total ionic conductivity of 5.09 × 10^−4^ S·cm^−1^, the relative density was 87.3%, and the smallest activation energy was 0.311 eV. For the quasi-solid-state batteries assembled with submicron LLZNO powder, the capacity retention rate was 99.15% and the specific capacity was 105.5 mAh/g after 50 cycles at room temperature with a current density of 0.02 mA/cm^2^. Therefore, we have presented a simple method to reduce the waste of raw materials and energy used when sintering LLZO ceramics. At the same time, the prepared submicron LLZO powder can also be applied in quasi-solid-state batteries, with a good electrochemical performance.

## Figures and Tables

**Figure 1 materials-13-00560-f001:**
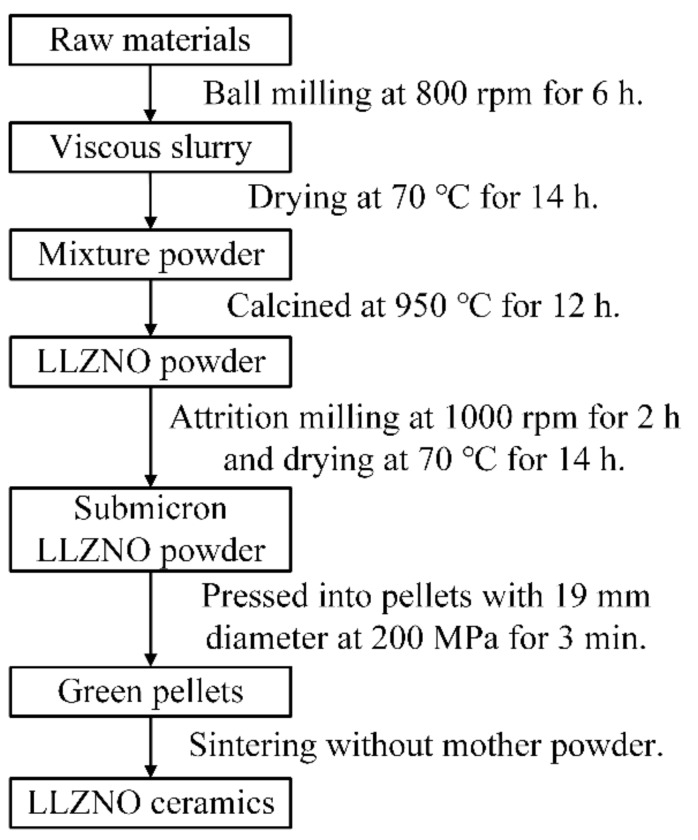
Process flow chart for the preparation of submicron Li_6.6_La_3_Zr_1.6_Nb_0.4_O_12_ (LLZNO) powder and the sintering of LLZNO ceramics.

**Figure 2 materials-13-00560-f002:**
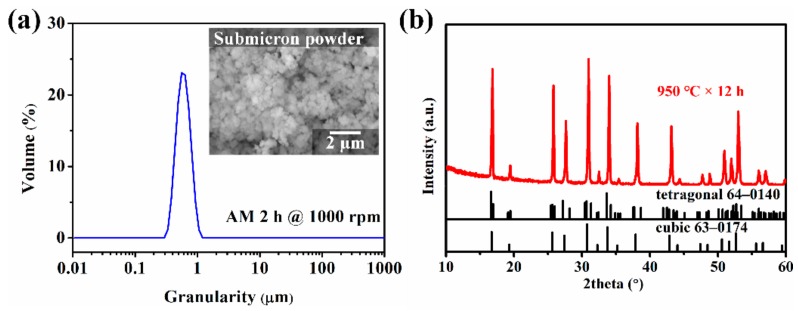
(**a**) Particle-size distribution of the LLZNO powder after being attrition milled 2 h at 1000 rpm and its SEM image and (**b**) XRD pattern of the LLZNO powder.

**Figure 3 materials-13-00560-f003:**
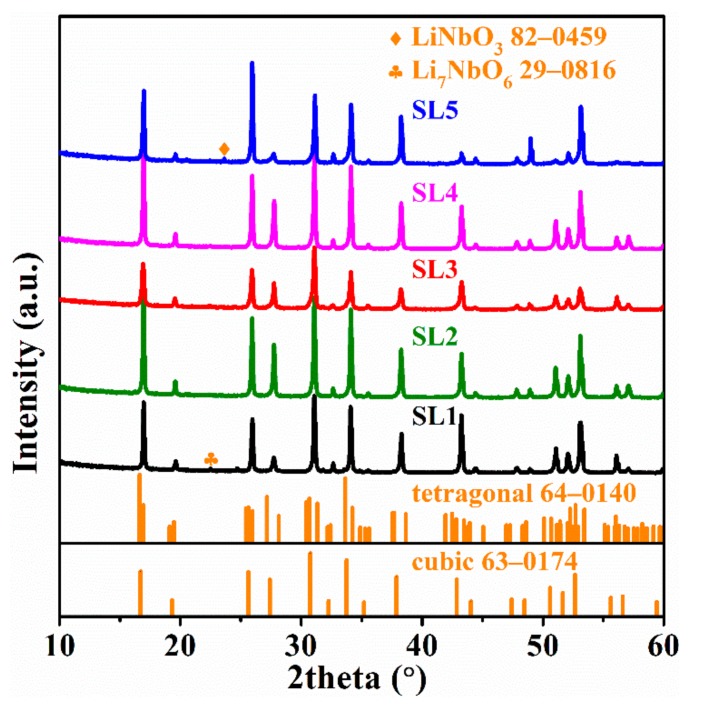
XRD patterns of the LLZNO ceramics with different sintering conditions.

**Figure 4 materials-13-00560-f004:**
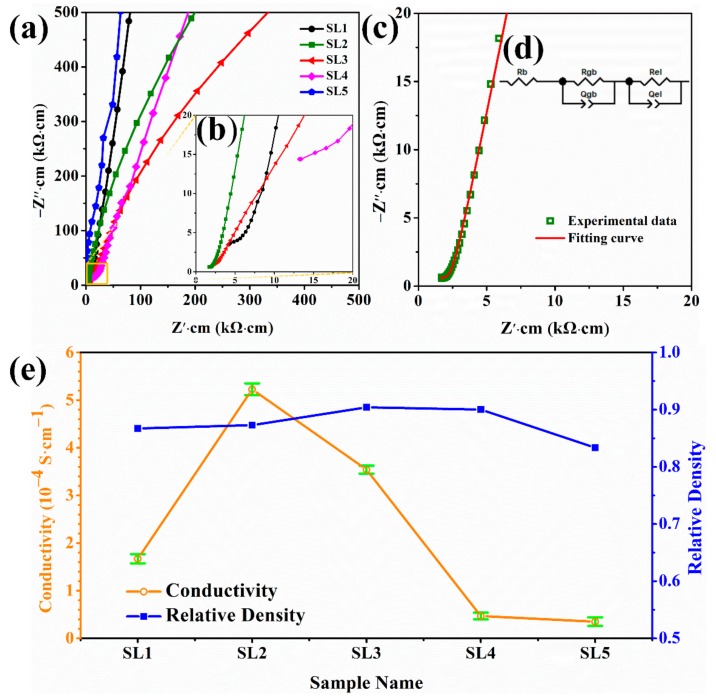
(**a**, **b**) AC impedance plots of the LLZNO ceramics with different sintering conditions at 25 °C; (**c**) AC impedance plots and fitting curve of SL2 at 25 °C; (**d**) equivalent circuit to fit the curves. (**e**) Total conductivity and relative density of the LLZNO ceramics.

**Figure 5 materials-13-00560-f005:**
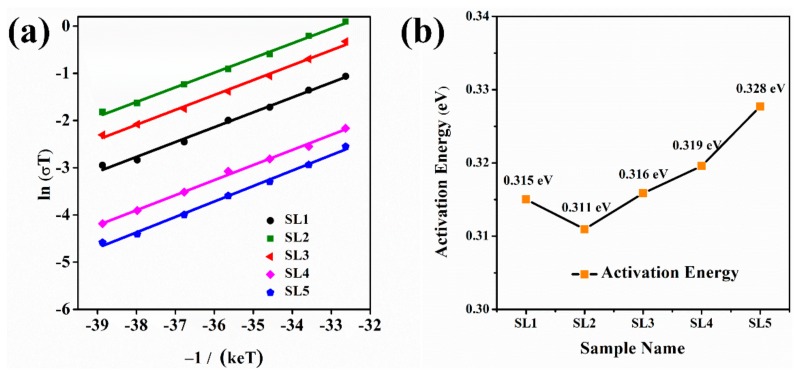
(**a**) Arrhenius plots and fitting results and (**b**) the activation energy of different LLZNO ceramics.

**Figure 6 materials-13-00560-f006:**
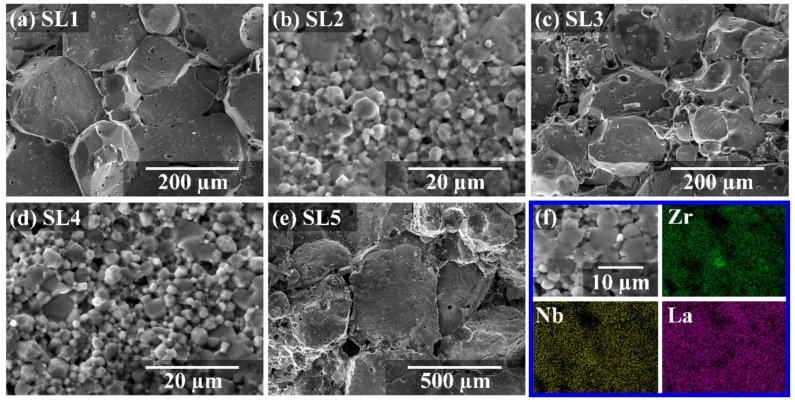
(**a**–**e**) SEM images of the cross-sectional microstructures of the ceramics that were sintered by different particles sizes under different sintering conditions and (**f**) EDS mapping of LLZNO ceramics section sintered at 1200 °C for 30 min.

**Figure 7 materials-13-00560-f007:**
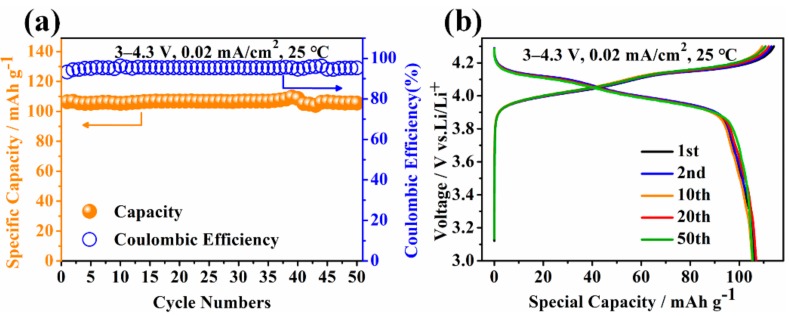
(**a**) Specific capacity and coulombic efficiency and (**b**) the 1st, 2nd, 10th, 20th, and 50th galvanostatic charge‒discharge curves of quasi-solid-state batteries with LiMn_2_O_4_ as the positive electrode.

**Table 1 materials-13-00560-t001:** Laser particle-size test results of submicron-scale LLZNO powder.

Preparation Condition	D_10_ (µm)	D_50_ (µm)	D_90_ (µm)	D_(3,2)_ (µm)	D_(4,3)_ (µm)	Specific Surface Area (m^2^/kg)
Attrition milled 2 h @ 1000 rpm	0.430	0.590	0.812	0.575	0.607	2007

**Table 2 materials-13-00560-t002:** Sintering condition, cell parameter, total ionic conductivity at 25 °C, activation energy, and relative density of LLZNO ceramics.

Sample Name	Sintering Condition	Cell Parameter (Å)	Total Ionic Conductivity (10^−4^ S·cm^−1^), 25 °C	Activation Energy (eV)	Relative Density
SL−1	1200 °C × 60 min	12.8952	1.58	0.315	86.7%
SL−2	1200 °C × 30 min	12.8953	5.09	0.311	87.3%
SL−3	1150 °C × 60 min	12.9028	3.49	0.316	90.4%
SL−4	1100 °C × 60 min	12.8916	0.51	0.319	90.3%
SL−5	1100 °C × 360 min	12.8870	0.35	0.328	83.4%

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
