# Peer review of "Submicron-Sized Nb-Doped Lithium Garnet for High Ionic Conductivity Solid Electrolyte and Performance of Quasi-Solid-State Lithium Battery"

_materials, 2020, doi:10.3390/ma13030560_

Round 1
Reviewer 1 Report
Revised manuscript dedicated to interesting scientific problem of the development advanced ceramic solid electrolytes with Li ion conductivity. Presented experimental results are quite interesting, however, are not coherent. Moreover, the manuscript in presented form has serious flaws which, from my modest point of view, make impossible its publishing in MATERIALS.
Below, authors can find more detailed comments.
General comment #1. Since liquid electrolyte was added to the investigated cell, it cannot be called as "all solid state cell", whereas I´d suggest more correct term: "quasi" or "pseudo solid state cell" in all text.
General comment #2. The manuscript has two parts (solid electrolyte synthesis and so-called "all solid state cell" which are not connected between each other. My recommendation is to remove the second "pseudo solid state cell" part from the manuscript.
Line 41. I think in this case it is better to use "increasing" instead of "improving".
Line 43-46. I do not understand the meaning of the sentence. Please clarify and add references too.
Line 48. I do not understand the meaning of the sentence. Please clarify.
Line 52. Please specify which metal cation. Titanium?
Line 102. I do not understand how "submicron LLZNO powder" was prepared. Please clarify.
Lines 101-103. More information about positive electrode (loading, density, current collector information etc.) and cell (format etc.) must be given.
Line 108. Composition of the liquid electrolyte must be described.
Line 165. I do not understand the meaning of the sentence. Please clarify.
Figure 4e. Please change header of X axis toward better clearance. To my criteria, "1150ºCx60min" point looks like an outlier. Please add error bar for all data set and explain better obtained results.
Table 2. I think authors should discuss contribution of bulk and grain boundary resistance for better explanation how sintering conditions affect the LLZNO pellet properties.
Figure 6. I cannot compare correctly presented SEM images. Please show ones with similar magnitude.
Reviewer 2 Report
This work is of a good scientific quality and can be recommended for publication after minor revision since some points should be clarified in the text.
1. The stability was evaluated for only 50 cycles that is very small number especially considering other works in the Li-ion batteries field where stability during thousands of cycles is demonstrated. Thus, the stability test in this work should be demonstrated for at least of several hundreds of cycles to evaluate material stability.
2. In introduction authors mentioned the conductivity of LLZO in the range of 10-3 S/cm however numbers demonstrated in the result part are order of magnitude higher. Please comment.
3. Line 108, authors mentioned that some amount of liquid organic electrolyte was also used. Please specify the composition of electrolyte.
4. XRD. Comparing with reference patterns some peak shift at high 2-Theta values was observed. It can be clearly seen for peaks at 50-60 degrees. Please comment.
5. Fig. 4e. I recommend to authors to re-arrange figure by switching on the x-axis first 2 points: to make most left point 1200Cx60min and then highest temperature shorter time (1200Cx30min) to be the 2nd one. Same with Fig. 5b.
6. The conductivity of the sample 1200Cx60mim is lower compared to 1150Cx60min (Fig. 5a) however, activation energy is also a bit lower (Fig. 5b). This point should be further discussed in the text.
7. Fig. 6f. In the central part of the map a segregation of Zr and Nb along with depletion of La can be seen. However, on the line 208 the sentence reads “the elements of La, Zr, and Nb are uniformly distributed”. This point should be discussed in the text in more details or image should be replaced with the other one.
Reviewer 3 Report
This manuscript, entitled "Submicron Sized Nb Doped Lithium Garnet for High Ionic Conductivity Solid Electrolyte and Performance of All Solid-State Lithium Battery", is considered to be relevant to the scope of this journal.
A good synthesis of the literature offering an overview of the evolution researches in the area.
In my opinion, for conductivity, relative density, activation energy, the calculation formulas or references should be specified.
A work of high scientific level with some minor changes to be made before publication.
Round 2
Reviewer 1 Report
The overall quality of the revised manuscript, mainly the first part (not include quasi-solid-state battery), has been slightly improved.
At the same time, I still do not see any sense to add the second (and small) part related to quasi-solid-state battery to the manuscript. Since, no evidence that LLZNO participates in the Li ion transport when it is wetted by a liquid electrolyte. Niether, ionic conductivity data of such quasi-solid electrolyte are presented. Moreover, in the response of authors they have just mentioned that "So, this part is necessary for the manuscript" but without solid scientific argumentation and/or other important considerations.
Therefore, the revised manuscript in presented form still has serious flaws which, from my modest point of view, make impossible its publishing in MATERIALS and therefore should be rejected.
